# The Risk of Malnutrition and Sarcopenia in Elderly People Living with HIV during the COVID-19 Pandemic

**DOI:** 10.3390/nu16152540

**Published:** 2024-08-02

**Authors:** Daylia Thet, Sawitee Lappichetpaiboon, Chidchanok Trakultritrung, Nongnapas Sotangkur, Supalak Phonphithak, Hay Mar Su Lwin, Tanakorn Apornpong, Win Min Han, Anchalee Avihingsanon, Tippawan Siritientong

**Affiliations:** 1Department of Food and Pharmaceutical Chemistry, Faculty of Pharmaceutical Sciences, Chulalongkorn University, Bangkok 10330, Thailand; daliathet@gmail.com (D.T.); tweety_oum@hotmail.com (S.L.); nanchidch@gmail.com (C.T.); nongnapasso.11@gmail.com (N.S.); 2The HIV Netherlands Australia Thailand Research Collaboration (HIV-NAT), Thai Red Cross AIDS Research Centre, Bangkok 10330, Thailand; supalak.k@hivnat.org (S.P.); hay.m@hivnat.org (H.M.S.L.); tanakorn.a@hivnat.org (T.A.); win.m@hivnat.org (W.M.H.); anchaleea2009@gmail.com (A.A.); 3Center of Excellence in Tuberculosis, Faculty of Medicine, Chulalongkorn University, Bangkok 10330, Thailand; 4Center of Excellence in Burn and Wound Care, Chulalongkorn University, Bangkok 10330, Thailand

**Keywords:** elderly, HIV, muscle mass, malnutrition, nutritional status, sarcopenia

## Abstract

Malnutrition is a risk factor of sarcopenia in the elderly. During the COVID-19 pandemic, limited transportation and supply chain disruptions restricted access to nutritious foods. We assessed the nutritional status and sarcopenia risk in older people living with HIV (PLWH) on combination antiretroviral therapy in Thailand. This study was a hospital-based cohort among virally suppressed older PLWH who came for routine HIV clinic visits. The mini nutritional assessment (MNA), body composition analysis and 5-time chair stand test (CST) were performed to assess the nutritional status, muscle mass and physical performance, respectively. A total of 177 PLWH were enrolled (57.60% male). The median age was 58 years (IQR 55–62 years). Thirty-five participants (19.8%) were at risk of developing sarcopenia, and 28.2% had abnormal nutritional status. Muscle mass correlated positively with nutritional scores (r = 0.355, *p* < 0.001) but negatively with 5-time CST duration (r = −0.173, *p* = 0.021). In the multivariate model, muscle mass was associated with age, sex, mid-arm circumference, calf circumference and 5-time CST duration. In a well-viral-suppressed older Asian PLWH cohort, given the positive correlation between nutritional status and muscle mass, the nutritional status of older PLWH should be routinely evaluated and monitored.

## 1. Introduction

During the pandemic of coronavirus disease 2019 (COVID-19), people living with human immunodeficiency virus infection (PLWH) had higher risk of complications and higher mortality rate compared to HIV-negative people [1]. The pandemic lockdown had a negative impact on the accessibility of nutritional support, eating habits, lifestyle behaviors and physical activity of the global population [2]. Meanwhile, in the HIV population, the increase in longevity leads to increased observations of many aging complications, including malnutrition, cognitive impairment and sarcopenia [3,4,5]. Thus, the nutritional status of older PLWH during the COVID-19 pandemic becomes critical. There are also growing concerns for food insecurity and nutrient deficiencies among older people with and without HIV due to pandemic restrictions.

The European Society of Parenteral and Enteral Nutrition (ESPEN) defines malnutrition as a condition whereby the patient does not consume enough food or has inadequate nutrition to meet the requirement of the body, characterized by changes in body composition, which lead to physical disability and impaired clinical outcomes [6,7]. In the Asia–Pacific region, a study from Taiwan reported that the prevalence of malnutrition and those at risk of malnutrition among older individuals were 2% and 13.1%, respectively, which highlighted the importance of nutritional care in this population [8]. Moreover, the nutritional status of older Asians living with HIV deteriorated over time [9], supporting the regular screening of malnutrition and sarcopenia in this group.

Poor nutritional status is a risk factor for sarcopenia [4,10]. The Asian Working Group of Sarcopenia (AWGS) defined sarcopenia as the age-associated loss of skeletal muscle mass in combination with lower muscle strength and poor physical performance. Aside from impaired muscle strength and function, with advanced age, their muscle mass continues to deteriorate, which in turn results in several complications such as poor physical performance, a decreased quality of life, hospitalization, and high morbidity and mortality [4,11]. Perhaps due to higher rates of comorbidities and ART-associated factors, the risk for low muscle mass is more common in older PLWH than in those without HIV [3]. The AWGS reported that the prevalence of sarcopenia ranged from 5.5% to 25.7%, and it was higher among males (5.1–21%) than females (4.1–16.3%) in the general population [12]. Using the AWGS criteria, a recent large-scale national study in Thailand reported that among the community-dwelling older adults, the overall sarcopenia prevalence was 18.1% while 66.7% had severe sarcopenia [13]. In Asian PLWH, the sarcopenia prevalence was 27.3% as defined by AWGS-2019 [14]. Currently, it is well recognized that bioelectrical impedance analysis (BIA) is a reliable method to assess the skeletal muscle mass to detect sarcopenia [13,15,16,17]. In addition, muscle mass can be preliminarily assessed by measuring the calf circumference [12], and physical performance can be evaluated by the 5-time chair stand test (CST), which was proposed as a diagnostic tool for sarcopenia [18]. The advantages of 5-time CST use during the COVID-19 pandemic are limited physical contact and the ability to conduct remote assessment, which typically helps limit contagious infection.

The nutritional requirement and body composition such as the muscle mass of the HIV population may also be highly affected by aging. However, there are limited data regarding the association of the nutritional status and muscle mass in Asian PLWH on cART. The spread of COVID-19 and the lockdown during the pandemic limited food accessibility and led PLWH to become more vulnerable to malnourishment. This study assessed the risk of malnutrition and sarcopenia in older PLWH on combination antiretroviral therapy (cART) in Thailand during the COVID-19 pandemic. The relationships between nutritional status and muscle mass, and potential factors affecting the decrease in muscle mass were also evaluated.

## 2. Materials and Methods

This hospital-based cohort was conducted in 2020 and 2021, during the COVID-19 outbreak, in older PLWH on cART at the HIV Netherlands Australia Thailand (HIV-NAT) Research Collaboration Centre, Bangkok, Thailand.

The study protocol was approved by the Institutional Review Board of the Faculty of Medicine, Chulalongkorn University. All participants provided written informed consent before they were enrolled into the study.

### 2.1. Study Population

The population of this study consisted of participants from a previously published study that assessed nutritional status and depression as primary outcomes [9]. Briefly, PLWH aged 50 years and above were consecutively recruited at the time of regular HIV care visits. PLWH who were unable to stand, had injury or were amputated in one of their limbs, or had implanted metal devices were excluded from the study. PLWH with Parkinson’s or Alzheimer’s disease or suspected to have neurological or cognitive impairments were also excluded.

We calculated the sample size based on the prevalence of those at risk of malnutrition (13.1%) [8] and precision (0.05); therefore, at least 175 PLWH were needed to assess the nutritional status and muscle mass in this study.

### 2.2. Study Assessments

#### 2.2.1. Nutritional Status

The Thai-version MNA was used to assess the nutritional status of PLWH in this study [9]. It is an 18-item questionnaire which includes (i) a general clinical assessment (mobility status, psychological stress, neuropsychological problems, living independently or in a nursing home, medications, skin ulcers); (ii) physical measurements (weight loss, BMI, mid-arm circumference, calf circumference); (iii) dietary assessment (food intake, number of meals, dietary intake, mode of feeding); and (iv) subjective questions (self-view of nutritional status, perception of health status). The results were categorized based on the following scores: normal nutritional status (24 to 30 points), at risk for malnutrition (17 to 23.5 points), and malnutrition (lower than 17 points).

#### 2.2.2. Body Composition

The BIA machine (BC-418, Tanita, Tokyo, Japan) was used to measure the body composition of PLWH such as muscle mass, fat mass, total body water and visceral fat rating. PLWH were asked to stand barefoot on the foot electrodes and hold the hand electrodes with their arms straight down, a few centimeters from their thighs. Mid-arm circumference, waist circumference, hip circumference and calf circumference were measured using the standard non-elastic tape. PLWH “at risk” for developing sarcopenia were defined as having a calf circumference of <34 cm in males and <33 cm in females, together with a duration of CST of more than 12 s. The appendicular skeletal mass index (ASMI) was also assessed. ASMI values of <7 kg/m^2^ (males) and <5.7 kg/m^2^ (females) were considered as low levels [12].

#### 2.2.3. Physical Performance

To assess the physical performance, the 5-time CST that is a validated distant assessment was applied. PLWH were asked to sit with their hands folded in front of their chests, without leaning back in their chair, under guidance by research staff. After being in that position, they were then requested to stand up as fast as possible five times. The time from the first sitting position to the final standing position were recorded. Low physical performance was defined as time taken to finish 5-time CST ≥12 s [12]. The participants were further inquired about physical exercise or movement beyond daily routines using yes/no questions.

#### 2.2.4. Sociodemographic Characteristics and HIV-Related Characteristics

The demographic and clinical data such as age, sex, marital status, occupation, duration of HIV infection, duration of cART, types of cART regimens, comorbidities and medical history were extracted from the electronic medical records. PLWH were asked about their exercise or physical activities, smoking and drinking habits. The most recent laboratory data including CD4, viral load, glucose, creatinine and lipid profiles were also collected.

### 2.3. Statistical Analysis

All data were analyzed using the Statistical Package for Social Sciences software (version 22). Distributions of the data were checked using the Kolmogorov–Smirnov test. Categorical variables were presented as frequency and percentage. The chi-squared test and Fisher’s exact test were used to compare the categorical variables, as appropriate. Continuous variables were presented as the median and the interquartile range (IQR). Pearson correlation and Mann–Whitney U test were used to compare the continuous variables. Multivariable linear regression analysis with robust standard errors was performed to assess the relationship of muscle mass and various clinical variables such as age, duration of HIV and cART, number of comorbidities and laboratory parameters. *p*-value < 0.05 was considered as statistically significant.

## 3. Results

The demographic and clinical characteristics of the PLWH included are presented in Table 1. One hundred and seventy-seven PLWH were enrolled. Most of the participants were male (57.6%) and the median age was 58 years. The median duration of ART use was 20 years. Ninety percent of PLWH were on triple ART regimens and a majority of them were on non-nucleoside reverse transcriptase inhibitor (NNRTI)-based cART (61.6%), followed by integrase strand transfer inhibitor (INSTI) (16.4%) and protease inhibitor (PI) (14.1%). Common comorbidities found were lipodystrophy (44.6%), hypertension (40.7%), diabetes (17.5%) and osteoporosis (3.4%). The median CD4 count was 633 cell/mm^3^, and 98.9% of PLWH had HIV-1 RNA <50 copies/mL. In our cohort, more than half of the participants (61.6%) had normal BMI (18.5 to <25 kg/m^2^), 32.2% had high BMI (≥25 kg/m^2^) and 6.2% had low BMI (<18.5 kg/m^2^).

### 3.1. Nutritional Status

Using MNA assessments, 127 PLWH (71.8%) had normal nutritional status, 46 participants (26%) were at risk of malnutrition, and 4 participants (2.2%) were malnourished. Overall, the median MNA score was 25.50 and males had significantly higher median MNA scores than females (26.00 vs. 24.50) (*p* = 0.024). PLWH with normal nutritional status had a median MNA score of 26.50, whereas those with abnormal nutritional status had a median MNA score of 22.00. Responses to each question of the MNA are mentioned in Appendix A.

### 3.2. Body Composition

Most of the anthropometric measurement data such as body weight, waist circumference, calf circumference, waist/hip ratio and muscle mass were significantly higher in males than females (*p* < 0.05). However, females had higher total fat mass and fat percentage than males.

PLWH with normal nutritional status had higher body weight, mid-arm circumference, waist circumference, hip circumference, calf circumference, waist/hip ratio, muscle mass, fat mass, visceral fat rating and total body water composition compared to those who were at risk of malnutrition or malnourished (*p* < 0.05).

### 3.3. Physical Performance

Fifty-five percent of PLWH reported that they had regular exercise activities. Using 5-time CST, 73 (41.2%) participants took ≥12 s to finish the test; 39 participants reported that they did not exercise regularly. The median time for 5-time CST between males and females were 10.00 s and 11.00 s, respectively. Among PLWH with good physical performance (5-time CST < 12 s), 61.5% exercised. We observed that exercise was associated with 5-time CST (*p* = 0.049).

### 3.4. Correlation and Association of Total Muscle Mass with Nutritional Status and Physical Performance

In the bivariate analysis, the nutritional scores were positively correlated with total muscle mass (*p* < 0.001). However, total muscle mass was negatively correlated with the duration of 5-time CST (*p* = 0.021) (Table 2).

The correlation of total muscle mass with nutritional scores, CST and age is presented in Figure 1. PLWH with good nutritional status were more likely to have greater total muscle mass and a positive correlation between total muscle mass and MNA scores. However, poor physical performance was positively correlated with lower total muscle mass in older PLWH.

In a bivariate regression analysis, age, sex, duration of CST, CD4 and HDL levels were negatively associated with muscle mass, whereas mid-arm circumference, calf circumference, glucose, hemoglobin levels and nutritional scores were positively associated. In multivariate model, age, sex, duration of CST, mid-arm circumference and calf circumference were independently associated with muscle mass (Table 3).

### 3.5. Prevalence of “At-Risk” Sarcopenia

We found that 35 (19.8%) PLWH were at risk for sarcopenia. Eleven participants had low ASMI values (Figure 2). Low ASMI was negatively correlated with age; therefore, older PLWH had lower ASMI (*p* = 0.001).

Table 4 shows that older patients were more likely to be sarcopenic (*p* = 0.033). The participants who were at risk of sarcopenia had significantly lower body weight, waist circumference, hip circumference, mid-arm circumference, calf circumference, muscle mass and total body water than those without sarcopenia risk (*p* < 0.01). PLWH with sarcopenia also had significantly higher fat mass than those without sarcopenia (*p* = 0.048).

## 4. Discussion

In this study, we assessed the association of malnutrition and sarcopenia in well-suppressed older PLWH with a median of 20 years on cART during the COVID-19 pandemic. Almost 30% of our older PLWH were at risk for malnutrition or malnourished and 20% were at risk for developing sarcopenia, using a calf circumference of <34 cm in males and <33 cm in females, together with a duration of CST >12 s. The nutritional scores (MNA) were also positively correlated with total muscle mass.

Our study can be comparable with other studies assessing the nutritional status and sarcopenia in patients with COVID-19 or other chronic diseases. In a recent report on non-HIV elderly patients with COVID-19 infection in China, 45% had sarcopenia, and of these, 35.3% were malnourished [19]. Interestingly, sarcopenia was found to be a risk factor for death in elderly people with COVID-19 infection in a Brazilian study [20]. Hospitalized elderly diagnosed with COVID-19 had a risk of sarcopenia (63.8%) and a risk of malnutrition (72%). Patients with sarcopenia were five times more likely to die, and likewise, malnutrition was a risk factor of death.

A study by Bahat et al. [21] reported that fat-free mass was significantly lower in malnourished patients than those with normal nutritional status. Moreover, the nutritional scores in our population were negatively correlated with the duration of 5-time CST, which indicated that participants with abnormal nutritional status may have poor physical performance. Likewise, in another study conducted in Peruvian community-dwelling older adults, abnormal nutritional status was associated with poorer physical performance [22]. There was a negative correlation between duration of 5-time CST with muscle mass, which indicated that the participants with low muscle mass were more likely to have poor physical performance. Our finding was in agreement with the previous study, which reported that there was a positive correlation between physical performance by 5-time CST and skeletal muscle mass [23].

It should be noted that muscle mass deteriorated with increasing age in our study. Age-related decline in muscle mass was seen in both healthy individuals [24] and aging PLWH [25,26]. In agreement with our finding, older people were more likely to be sarcopenic based on a previous report in China [19]. Similarly, in hypertensive elderly patients with COVID-19 infection in China, older people aged 80 years and above were more likely to suffer from sarcopenia (39.8% vs. 19.5%) than the younger group [27].

In our cohort, females were more likely to have lower nutritional scores compared to males. Moreover, females had lower calf circumference than the reference value [12] but they had higher waist circumference than the recommended values (≤80 cm) [28]. We also found that median total muscle mass was significantly greater in males than females. Our results are comparable with a recent report on Thai HIV-infected individuals aged 20–75 years, in which the prevalence of sarcopenia was 21.9% [17]. Older age, female gender, a longer duration of HIV infection, and a longer duration of ART were reported to be risk factors for sarcopenia. Janssen et al. [24] reported that there were sex differences in the distribution of skeletal muscle mass in healthy individuals. On average, males had 36% greater skeletal muscle mass than females. In contrast, male PLWH were more likely to be at risk for low muscle mass compared to the females [25,26].

In our cohort, we also observed that BMI was positively correlated with muscle mass. The finding was aligned with a previous report on PLWH. Patients with higher BMI had less likelihood of low muscle mass (OR: 0.43, 95% CI: 0.34–0.56) [16]. Likewise, our result was in line with the previous finding that BMI was a predictor for low skeletal muscle mass in a healthy older population [29]. In participants with abnormal nutritional status, the median calf circumference was significantly lower than the values specified in the assessment criteria for sarcopenia according to AWGS-2019 [12]. It is possible that malnourished patients with low muscle mass may be at risk of developing sarcopenia.

We found that nearly one-fifth of our participants were at risk for sarcopenia, of which the majority were females. Comorbidities such as lipodystrophy (48.4%) and hypertension (51.4%) were observed. Older participants were more likely to be sarcopenic, had lower muscle mass, lower nutritional scores and lower ASMI. The negative association of age and sarcopenia is supported by a previous Asian PLWH study conducted in Hong Kong (*p* < 0.001) [14]. However, the study did not find a gender contribution despite the majority of patients being male. In a previous study conducted in a nursing home for elderly residents, the risk of sarcopenia was 18.4% and females were more likely to be sarcopenic than the male residents [30]. In contrast, the risk of sarcopenia was higher in older males than older females (68% vs. 21%) in a nursing home in Italy [31]. Moreover, they highlighted the association of muscle mass with nutritional status and physical activity. The differences in the risk for developing sarcopenia and the association of muscle mass with various variables may be due to different diagnosis algorithms used in the studies from other regions of the world to detect sarcopenia or low-muscle-mass assessment.

Wu et al. [15] reported that women had low MNA scores and poor physical performance, which were associated with sarcopenia. In a previous study, female participants had lower muscle mass, and they were at higher risk for developing sarcopenia compared to males in the study. In contrast, a study conducted in a community-dwelling elderly population showed that males were more likely to be sarcopenic than females [32]. Although the study was conducted during the COVID-19 pandemic and the restrictions, it did not affect overall health and HIV suppression rates in the cohort, where most of the PLWH (98%) in our study had good control of HIV, high CD4 cell counts and normal albumin, hemoglobin, and creatinine levels. However, the median LDL cholesterol of PLWH was high in our study population.

There were some limitations in our study. During the COVID-19 pandemic period, health authorities suggested that people at high risk (i.e., elderly population) of acquiring COVID-19 infection should stay at home. Therefore, we were unable to enroll more PLWH who were older than 60 years, and patients might have had limited access to nutritious food, together with limited social and psychological support during the pandemic. Second, it is possible that there were recall biases regarding exercise. Because exercise was self-reported, we did not classify the category and duration of exercise. Third, due to the cross-sectional design of our study, we could not establish the causality and the estimates could be affected by other uncontrolled confounders. Fourth, our study was conducted only in Thai PLWH older than 50 years; the findings may therefore not be generalizable to other older PLWH whose lifestyles, food habits and geographics are different. The findings from this study showed that poor nutritional status, low muscle mass and low physical activity are important risk factors for developing sarcopenia in aging PLWH. Therefore, physicians should encourage older PLWH on ART to maintain good nutritional status, which will help prevent muscle mass loss. Moreover, temporal changes in the risk of sarcopenia should be assessed to better understand the impact of COVID-19 in elderly PLWH. The findings of this study support the need for a plan to diminish these complications in future events that may be similar to the COVID-19 outbreak.

## 5. Conclusions

In a well-viral-suppressed older Asian PLWH cohort, abnormal nutritional status was relatively high at 28%, and approximately 20% were at risk of developing sarcopenia. Given the positive correlation of nutritional status with muscle mass, the nutritional status of older PLWH should be routinely evaluated at HIV care services. Interventions to prevent age-related loss of muscle mass in older PLWH should also be undertaken to prevent future sarcopenia-related morbidity and mortality.

## Figures and Tables

**Figure 1 nutrients-16-02540-f001:**
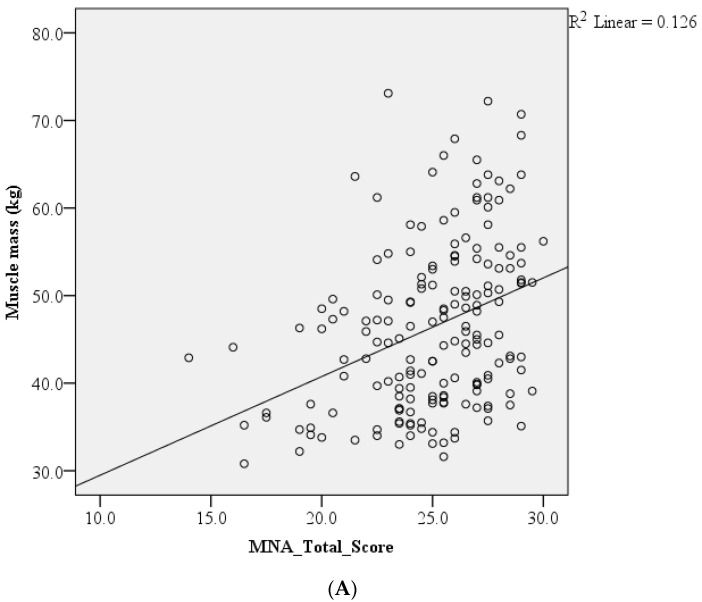
Scatter plots for the correlation of total muscle mass with (**A**) nutritional scores, (**B**) chair stand test and (**C**) age.

**Figure 2 nutrients-16-02540-f002:**
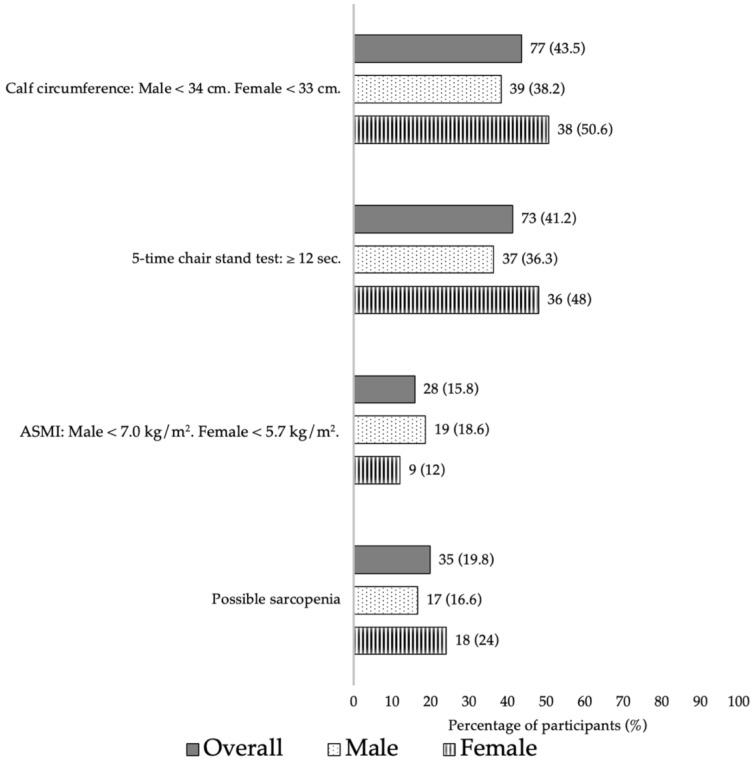
Proportion of possible sarcopenia and its components.

**Table 1 nutrients-16-02540-t001:** Characteristics of participants (*n* = 177).

Characteristics	Total	Male (*n* = 102)	Female (*n* = 75)	*p*-Value *
Age (years), median (IQR)	58.00 (55.00–62.00)	58.00 (55.00–61.25)	58.00 (56.00–62.00)	0.301
Marital status, n (%)				<0.001
Single	42 (23.7)	36 (35.3)	6 (8)	
Married	53 (29.9)	36 (35.3)	17 (22.7)	
Other	82 (46.4)	30 (29.4)	52 (69.3)	
Occupation, n (%)				0.046
Employed	140 (79.1)	86 (84.3)	54 (72)	
Unemployed	37 (20.9)	16 (15.7)	21 (28)	
Drinking, n (%)				0.001
Yes	36 (20.3)	29 (28.4)	7 (9.3)	
No	141 (79.7)	73 (71.6)	68 (90.7)	
Smoking, n (%)				0.001
Yes	18 (10.2)	17 (16.7)	1 (1.3)	
No	159 (89.8)	85 (83.3)	74 (98.7)	
Exercise, n (%)				0.641
Yes	98 (55.4)	58 (56.9)	40 (53.3)	
No	79 (44.6)	44 (43.1)	35 (46.7)	
Anthropometric data, median (IQR)				
Height (cm)	163.00 (155.50–170.00)	168.00 (165.00–171.25)	155.00 (150.00–159.00)	<0.001
Body weight (kg)	60.40 (53.80–68.35)	65.05 (57.97–73.20)	56.80 (50.10–60.40)	<0.001
Mid-arm circumference (cm)	27.00 (25.00–28.00)	27.00 (25.00–28.00)	26.50 (25.00–28.50)	0.733
Calf circumference (cm)	33.50 (31.00–36.00)	34.50 (32.95–36.62)	32.00 (30.00–35.00)	<0.001
Waist circumference (cm)	86.00 (80.00–92.00)	88.00 (81.00–93.25)	84.00 (78.00–90.00)	0.039
Hip circumference (cm)	94.00 (89.00–98.00)	94.00 (89.00–98.00)	95.00 (89.00–99.00)	0.479
Waist/hip ratio	0.92 (0.86–0.96)	0.94 (0.88–0.97)	0.88 (0.84–0.94)	<0.001
Muscle mass (kg)	45.10 (38.30–53.05)	51.25 (47.17–56.92)	37.70 (35.20–40.60)	<0.001
Fat mass (kg)	15.20 (11.30–19.55)	12.95 (9.90–18.02)	17.80 (14.70–21.30)	<0.001
Fat (%)	25.10 (19.25–31.80)	19.90 (16.75–25.10)	32.10 (27.50–35.20)	<0.001
Visceral fat rating	9.00 (6.50–11.00)	10.00 (8.00–12.00)	7.00 (6.00–8.00)	<0.001
Total body water (kg)	33.00 (28.05–38.85)	37.55 (34.57–41.65)	27.60 (25.80–29.70)	<0.001
Body mass index (kg/m2)	23.30 (20.80–25.65)	23.25 (20.80–25.52)	22.00 (18.00–26.00)	0.778
Duration of HIV infection (years), median (IQR)	23.00 (19.00–25.00)	23.00 (19.00–25.00)	22.00 (18.00–26.00)	0.878
Duration of ART (years), median (IQR)	20.00 (16.00–23.00)	20.00 (15.75–23.00)	20.00 (16.00–23.00)	0.927
ART regimens, n (%)				0.100
NNRTI-based regimen	109 (61.6)	59 (57.8)	50 (66.7)	
INSTI-based regimen	33 (18.6)	17 (16.7)	16 (21.3)	
PI-based regimen	31 (17.5)	22 (21.6)	9 (12)	
Other	4 (2.3)	4 (3.9)	0 (0)	
Number of comorbidities, median (IQR)	4.00 (2.00–5.00)	4.00 (2.00–5.00)	4.00 (3.00–5.00)	0.453
Types of comorbidities				
Hypertension	72 (40.7)	48 (47.1)	24 (32)	0.044
Diabetes	31 (17.5)	25 (24.5)	6 (8)	0.005
Lipodystrophy	79 (44.6)	45 (44.1)	34 (45.3)	0.872
Osteoporosis	6 (3.4)	1 (1)	5 (6.7)	0.084
Laboratory data, median (IQR)				
CD4 (cells/mm3)	633.00 (474.50–777.50)	574.50 (420.75–741.25)	685.00 (540.00–840.00)	<0.001
Albumin (g/dL)	4.40 (4.08–4.60)	4.40 (4.10–4.70)	4.30 (4.00–4.50)	0.059
Hemoglobin (g/dL)	14.10 (13.00–15.00)	14.70 (13.87–15.80)	13.30 (12.60–14.10)	<0.001
Creatinine (mg/dL)	0.93 (0.78–1.08)	1.01 (0.93–1.15)	0.78 (0.70–0.87)	<0.001
AST (U/L)	26.00 (21.00–30.50)	27.00 (21.00–31.00)	24.00 (20.00–30.00)	0.083
ALT (U/L)	27.00 (20.50–38.00)	28.50 (22.00–42.00)	26.00 (18.00–34.00)	0.059
hs-CRP (mg/L)	0.10 (0.05–0.25)	0.09 (0.05–0.24)	0.11 (0.05–0.25)	0.608
Insulin (μU/mL)	6.90 (4.65–10.40)	6.70 (4.47–10.00)	7.20 (5.00–11.70)	0.289
Glucose (mg/dL)	96.00 (87.00–106.00)	98.50 (91.00–112.25)	92.00 (85.00–101.00)	<0.001
Triglycerides (mg/dL)	120.00 (84.00–176.50)	129.00 (84.00–193.50)	111.00 (80.00–149.00)	0.014
LDL (mg/dL)	126.50 (100.75–127.00)	122.50 (95.87–143.15)	134.35 (108.45–154.25)	0.091
HDL (mg/dL)	49.00 (40.00–58.00)	44.00 (38.00–55.00)	54.00 (45.00–63.00)	<0.001
Duration of CST (second), median (IQR)	11.00 (9.00–13.00)	10.00 (8.00–13.00)	11.00 (9.00–14.00)	0.027
MNA scores, median (IQR)	25.50 (23.50–27.00)	26.00 (24.00–27.50)	24.50 (23.50–27.00)	0.024
Nutritional status, n (%)				
Normal nutritional status	127 (71.8)	79 (77.4)	48 (64)	0.143
At risk of malnutrition	46 (26)	21 (20.6)	25 (33.3)	
Malnourished	4 (2.2)	2 (2)	2 (2.7)	

ALT, alanine aminotransferase; ART, antiretroviral therapy; AST, aspartate aminotransferase; CST, chair stand test; HDL, high-density lipoprotein; HIV, human immunodeficiency virus; LDL, low-density lipoprotein; MNA, mini nutritional assessment; NNRTI, non-nucleoside reverse transcriptase inhibitor; INSTI, integrase strand transfer inhibitor; IQR, interquartile range; PI, protease inhibitor. * Mann–Whitney U test was used to compare the median between genders.

**Table 2 nutrients-16-02540-t002:** Correlation between nutritional scores, muscle mass, CST and some clinical variables.

	Nutritional Scores	Muscle Mass	Duration of 5-Time CST
Age (years)	−0.155 ^a^	−0.241 ^b^	0.102
BMI (kg/m^2^)	0.491 ^c^	0.438 ^c^	0.056
Duration of HIV (years)	−0.135	0.019	−0.202 ^b^
Duration of ART (years)	−0.139	−0.051	−0.146
Number of comorbidities	−0.207 ^b^	−0.084	−0.003
CD4 (cells/mm^3^)	0.070	−0.187 ^a^	0.086
Nutritional scores	-	0.355 c	−0.061
Muscle mass (kg)	0.355 ^c^	-	−0.173 ^a^
Duration of 5-time CST (second)	−0.061	−0.173 ^a^	-

ART, antiretroviral therapy; BMI, body mass index; CST, chair stand test; HIV, human immunodeficiency virus. ^a^ *p* < 0.05, ^b^ *p* < 0.01, ^c^ *p* < 0.001.

**Table 3 nutrients-16-02540-t003:** Associated factors of total muscle mass by multivariate linear regression analysis.

	Bivariable Analysis				Multivariable Analysis			
	Coefficient	95% Confidence Interval		*p*-Value	Coefficient	95% Confidence Interval		*p*-Value
Age (years)	−0.45	−0.71	−0.18	0.001	−0.17	−0.28	−0.07	0.002
Sex (female vs. male)	−14.41	−16.32	−12.50	<0.001	−11.16	−12.62	−9.70	<0.001
Calf circumference (cm)	2.19	1.88	2.51	<0.001	1.1	0.81	1.38	<0.001
Mid-arm circumference (cm)	1.46	1.05	1.87	<0.001	0.73	0.43	1.02	<0.001
Duration of 5-time CST (sec)	−0.49	−0.91	−0.08	0.021	−0.18	−0.34	−0.03	0.015
CD4 by 100 (cells/mm^3^)	−0.76	−1.35	−0.16	0.013	−0.01	−0.22	0.20	0.921
Glucose (mg/dL)	0.07	0.03	0.12	0.002	0.01	−0.02	0.03	0.600
HDL (mg/dL)	−0.24	−0.34	−0.14	<0.001	−0.02	−0.07	0.03	0.417
Hemoglobin (g/dL)	2.14	1.28	3.01	<0.001	0.05	−0.46	0.57	0.841
Nutritional scores	1.13	0.68	1.57	<0.001	−0.14	−0.37	0.10	0.248

**Table 4 nutrients-16-02540-t004:** Comparison of the anthropometric characteristics between participants with or without sarcopenia.

Characteristics	Sarcopenic (*n* = 35)	Non-Sarcopenic (*n* = 142)	*p*-Value
Sex, n (%)			0.255
Male	17 (48.6)	85 (59.9)	
Female	18 (51.4)	57 (40.1)	
Age (years), median (IQR)	59 (57.00–63.00)	57 (55.00–62.00)	0.033
BMI (kg/m^2^), median (IQR)	20.80 (19.40–23.30)	23.70 (21.50–25.80)	<0.001
Exercise, n (%)			0.448
Yes	17 (48.6)	81 (57)	
No	18 (51.4)	61 (43)	
Diabetes, n (%)	7 (20)	24 (16.9)	0.628
Hypertension, n (%)	13 (37.1)	59 (41.5)	0.703
Lipodystrophy, n (%)	17 (48.6)	62 (43.7)	0.705
Duration of HIV infection (years), median (IQR)	22 (18–25)	23 (19–25)	0.664
Duration of ART (years), median (IQR)	20 (17.00–23.00)	20 (15.75–23.00)	0.718
CD4 (cells/mm^3^), median (IQR)	675 (484.00–851.00)	623 (470.50–776.75)	0.668
Nutritional scores, median (IQR)	25.00 (22.50–27.00)	25.50 (23.50–27.50)	0.050
Nutritional status, n (%)			0.096
Normal nutritional status	21 (60)	106 (74.6)	
Abnormal nutritional status	14 (40)	36 (25.4)	
Body weight (kg)	54.70 (49.20–60.40)	62.55 (56.52–70.52)	<0.001
Waist circumference (cm)	84.00 (75.00–87.00)	87.50 (81.00–94.00)	0.005
Hip circumference (cm)	89.00 (88.00–95.00)	95.00 (90.00–99.00)	<0.001
Mid-arm circumference (cm)	25.00 (23.00–27.00)	27.00 (25.00–29.00)	<0.001
Calf circumference (cm)	31.00 (30.00–32.50)	34.25 (32.50–36.00)	<0.001
Fat %	23.70 (19.30–32.10)	25.60 (19.12–31.80)	0.879
Fat mass (kg)	13.00 (10.10–17.00)	16.30 (11.80–19.85)	0.048
Muscle mass (kg)	37.70 (35.20–47.10)	46.75 (39.77–54.52)	<0.001
Total body water (kg)	27.60 (25.80–34.50)	34.20 (29.10–39.92)	<0.001

## Data Availability

The datasets generated during and/or analyzed during the current study are available from the corresponding author on reasonable request. The data are not publicly available due to privacy concerns of the vulnerable population.

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
