# Peer review of "The Risk of Malnutrition and Sarcopenia in Elderly People Living with HIV during the COVID-19 Pandemic"

_nutrients, 2024, doi:10.3390/nu16152540_

Round 1

Reviewer 1 Report

Comments and Suggestions for Authors

This study provides valuable information. However, the following points are unclear. Further revision is necessary before publication.

1. What is the most important originality in this study?

2. What is the most important impact of this study?

3. How about the comparison on the malnutrition and sarcopenia in the hospital-based cohort conducted before and during the COVID-19 outbreak, at the HIV-NAT Research Collaboration Centre, Bangkok, Thailand? 

4. How about the study on the malnutrition and sarcopenia in other countries during the COVID-19 outbreak?

5. Overall, did this study show little gender differences in the risk of malnutrition and sarcopenia in elderly people living with HIV during the COVID-19 pandemic? 

6. Is this the first study to investigate the status of the malnutrition and sarcopenia in elderly during the COVID-19 outbreak? 

Author Response

This study provides valuable information. However, the following points are unclear. Further revision is necessary before publication.

  1. What is the most important originality in this study?

Authors’ reply: We appreciate the reviewer’s question. Due to the increasing elderly population, various complications have arisen, including malnutrition and sarcopenia. These complications may have been overlooked during the COVID-19 pandemic, especially elderly PLWH.

The most important originality of this study is that we investigated malnutrition and sarcopenia in the elderly PLWH on cART during the COVID-19 pandemic in Thailand. The intersection of muscle mass and nutritional status makes our study among the first amidst the global health crisis. We can provide new insight on this specific concern. We performed comprehensive analysis by considering factors such as duration of HIV, duration of antiretroviral treatment, other comorbid conditions. Even though being the cross-sectional study cannot fill the gap of causality, we are confident that our data can provide fundamental benefit for targeted nutritional support and HIV healthcare strategies.

  1. What is the most important impact of this study?

Authors’ reply: Thank you for your feedback. This study highlights the occurrence of malnutrition and sarcopenia among the elderly PLWH on cART during the COVID-19 pandemic under restricted transportation conditions and limited access to quality food, together with physical and psychological complications. This vulnerable group may experience a lot of strikes such as being aging which makes more nutrient needs, impact of treatment and HIV infection in addition to COVID-19 lock down.

Although majority of patients in our study are virally well-suppressed, they still are at risk of malnutrition and sarcopenia. The findings of this study support the need for plan development to diminish these complications in future event that may be similar to COVID-19 outbreak. Individually targeted nutritional interventions or monitoring of muscle function may mitigate the potential negative outcomes in HIV care during the crisis. Moreover, to date, as the studies on malnutrition and sarcopenia risk in the elderly PLWH are limited, we can add clinical evidence to highlight the importance of routine evaluation of these problems, and to participate in future protocols. In addition, further studies can imply our findings to extend research assessing quality of life or causality in PLWH with poor nutritional status, sarcopenia and poor physical status.

  1. How about the comparison on the malnutrition and sarcopenia in the hospital-based cohort conducted before and during the COVID-19 outbreak, at the HIV-NAT Research Collaboration Centre, Bangkok, Thailand?

Authors’ reply: We do appreciate your suggestion. Comparison on the malnutrition and sarcopenia in the hospital-based cohort before and during the COVID-19 outbreak in our setting would provide valuable insights. In our previous publication by Thet et al. [ref no.9], we analysed the changes in nutritional status of the elderly PLWH over time: between baseline (2016-2017) and follow-up (2020-2021, COVID-19 pandemic period) at the HIV-NAT. The proportion of patients with malnutrition were significantly higher at the follow-up than those at the baseline. Pre-pandemic data of sarcopenia in our population were not available, unfortunately. However, in this study, we added risk of sarcopenia as our primary outcome. We acknowledge your comment and would like to recommend future studies to address this aspect.

We mentioned “Moreover, temporal changes in the risk of sarcopenia should be assessed to better understand the impact of COVID-19 in elderly PLWH.” in the discussion section. [PAGE 12, line 318-319]

  1. How about the study on the malnutrition and sarcopenia in other countries during the COVID-19 outbreak?

Authors’ reply: Thank you for your insightful question. There are reports on the malnutrition and/ or sarcopenia in other countries during the COVID-19 outbreak, of which, nevertheless, are conducted in non-HIV population. The reports primarily pointed out the negative impact of sarcopenia on clinical outcome. One common finding we can draw is that the more advanced the age, the higher the risk of sarcopenia. Yet, the studies particularly focusing on the elderly PLWH is scarce. In the manuscript, we added some literatures focusing on malnutrition/ sarcopenia as the followings:

“Our study can be comparable with other studies assessing the nutritional status and sarcopenia in patients with COVID-19 or other chronic diseases. In a recent report on non-HIV elderly patients with COVID-19 infection in China, 45% had sarcopenia, and of these, 35.3% were malnourished [19]. Interestingly, sarcopenia was found to be a risk factor for death in elderly people with COVID-19 infection in a Brazilian study [20]. Hospitalized elderly diagnosed with COVID-19 had a risk of sarcopenia (63.8%) and a risk of malnutrition (72%). Patients with sarcopenia were five times more likely to die, likewise, malnutrition was a risk factor of death. [PAGE 10, line 236-243]

In agreement with our finding, older people were more likely to be sarcopenic in previous report in China [19]. Similarly, in hypertensive elderly patients with COVID-19 infection in China, older people aged 80 years and above were more likely to suffer from sarcopenia (39.8% vs 19.5%) than the younger group [27].” [PAGE 10, line 257-260]

Ref 19: Zong, M., Zhao, A., Han, W., Chen, Y., Weng, T., Li, S., Tang, L., & Wu, J. (2024). Sarcopenia, sarcopenic obesity and the clinical outcome of the older inpatients with COVID-19 infection: a prospective observational study. BMC geriatrics, 24(1), 578. https://doi.org/10.1186/s12877-024-05177-w

Ref 20: Vasconcelos LGL, Me Mpomo JSV de M, Macena M, Souza TOM de, Dias C de A, Vasconcelos SML, et al. Sarcopenia and risk of malnutrition as risk factors for complications from COVID-19. Medicina (Ribeirão) 2023;56(3):e-206364. https://doi.org/10.11606/issn.2176-7262.rmrp.2023.206364

Ref 27: Xu, Q., Li, F., & Chen, X. (2023). Factors Affecting Mortality in Elderly Hypertensive Hospitalized Patients with COVID-19: A Retrospective Study. Clinical interventions in aging, 18, 1905–1921. https://doi.org/10.2147/CIA.S431271

  1. Overall, did this study show little gender differences in the risk of malnutrition and sarcopenia in elderly people living with HIV during the COVID-19 pandemic?

Authors’ reply: Thank you for your thoughtful comment on gender differences. We analyzed the potential differences in the risk of malnutrition and sarcopenia between genders among the elderly PLWH. Nonetheless, gender did not appear to be a major contributing factor for malnutrition and sarcopenia in our patients (p > 0.05) although there are differences in prevalences.

As we have described in Table 4, gender differences in risk of sarcopenia do not reach statistical significance (p = 0.255). Also, in terms of malnutrition, there is no statistically significant differences between male and female (Table 1).

  1. Is this the first study to investigate the status of the malnutrition and sarcopenia in elderly during the COVID-19 outbreak?

Authors’ reply: We appreciate your question regarding the novelty of our study. Specifically, this study investigated malnutrition and sarcopenia in the elderly PLWH during the COVID-19 outbreak in Thailand. While there have been studies on malnutrition and/ or sarcopenia the general (non-HIV elderly) population during this pandemic, original research focusing on the elderly PLWH is remarkably scarce. To our best knowledge, this is the first study in this vulnerable population in our country during the COVID-19 pandemic.

Reviewer 2 Report

Comments and Suggestions for Authors

This is a statistical study regarding the nutritional status and sarcopenia in people with HIV disease during the COVID pandemic. Although the statistical methods are adequate, the results and conclusions of the study are not reliable as there are many confounding factors (i.e., geographic residence of the subjects related to food availability, type of nutrition, the unreliability of the self-reported exercise, psychological status and the degree of loneliness, unaccounted comorbidities). Although the study design is adequate, all these confounding factors should be normalized in order to draw reliable conclusions regarding the impact of COVID pandemic on the physical status of the sublects.

Author Response

This is a statistical study regarding the nutritional status and sarcopenia in people with HIV disease during the COVID pandemic. Although the statistical methods are adequate, the results and conclusions of the study are not reliable as there are many confounding factors (i.e., geographic residence of the subjects related to food availability, type of nutrition, the unreliability of the self-reported exercise, psychological status and the degree of loneliness, unaccounted comorbidities). Although the study design is adequate, all these confounding factors should be normalized in order to draw reliable conclusions regarding the impact of COVID pandemic on the physical status of the subjects.

Authors’ reply: We appreciate the reviewer’s comments on our manuscript in terms of the impact of confounding factors on the reliability of our findings. As we have mentioned in our limitations, the results may be impacted by some confounders in our study. However, to tackle the effects of confounding factors, we applied multivariable regression to adjust the potential effects of multiple factors by putting simultaneously in the regression model. On the other hand, conducting in homogenous sample i.e., elderly Thai PLWH by some means reduces the variability as the data curation is consistent. We would like to confirm that the reliable statistical methods are taken to avoid the effects of confounders as much as possible by using multiple regression analysis. To ensure validity, we also have taken some procedures such as checking multicollinearity or assessing fitness of model.

For self-reported exercise, we have described this factor as one of limitations in our study as we were not able to identify type or duration of exercise.

We added in the discussion section as follow.

“…patients might have limited access to nutritious food together with limited social and psychological support during the pandemic. [PAGE 11, line 306-308]

The findings of this study support the needs of plan development to diminish these complications in future event that may be similar to COVID-19 outbreak.”. [PAGE 11, line 308-310]

Round 2

Reviewer 2 Report

Comments and Suggestions for Authors

The Authors still have not addressed the criticism raised in the previous version of the review. Indeed, they applied multivariate statistical analysis to minimize the impact of the confounding factors but nevertheless, such factors as those mentioned in the previous version of the text should have been accounted for in the first place, as they were available at the time of the study in the medical records of the patients.

Author Response

Comment:

The Authors still have not addressed the criticism raised in the previous version of the review. Indeed, they applied multivariate statistical analysis to minimize the impact of the confounding factors but nevertheless, such factors as those mentioned in the previous version of the text should have been accounted for in the first place, as they were available at the time of the study in the medical records of the patients.

Authors’ reply: Thank you very much for your valuable feedback. We do apologize for not fully addressing the concerns raised in the previous version of the review. We respectfully would like to try our best to reply to your comment.

We recognize that geographic residence, type of nutrition, unreliability of self-reported exercise, and other unaccounted comorbidities are critical considerations. Unfortunately, these data were not available in the medical records during the COVID-19 situation. However, to assess nutritional status, we used the Mini Nutritional Assessment (MNA). This 18-item questionnaire comprehensively assesses various factors including general clinical assessment (mobility status, psychological stress, neuropsychological problems, living conditions, medications, skin ulcers), physical measurements (weight loss, BMI, mid-arm circumference, calf circumference), dietary assessment (food intake, number of meals, dietary intake, mode of feeding), and subjective questions (self-view of nutritional status, perception of health status).

Given the numerous factors involved in assessing nutritional status, some of the parameters you mentioned were partially addressed. For example, 4 patients with normal nutritional status who lived in nursing homes as mentioned in Supplementary Material 1 (item G). MNA question items D and E focused on the psychological stress and neuropsychological problems, respectively, which were significantly associated with the nutritional status. In this study, our objective was to assess overall nutritional status and sarcopenia risk, predominantly. Due to the many restrictions during the pandemic, we faced constraints in the duration of patient interviews, which limits the depth of our data collection. Regrettably, we did not anticipate unaccounted comorbidities. Despite these limitations, we aimed to provide a thorough assessment through this well-validated questionnaire and statistical analysis to account for as many confounding factors as possible. We appreciate your understanding of the challenges faced during this period and hope that the comprehensive nature of our questionnaire provides a robust assessment of the subjects despite these constraints.

We mentioned the detailed components of the MNA under the section 2.2.1. Nutritional status as the followings:

“It is an 18-item questionnaire which includes (i) general clinical assessment (mobility status, psychological stress, neuropsychological problems, living independently or in nursing home, medications, skin ulcers), (ii) physical measurements (weight loss, BMI, mid-arm circumference, calf circumference), (iii) dietary assessment (food intake, number of meals, dietary intake, mode of feeding), and (iv) subjective questions (self-view of nutritional status, perception of health status).” [PAGE 3, line 103-108]

Additionally, we present separate analyses of responses to each question in the MNA between the normal nutritional status group and abnormal nutritional status group in Supplementary material 1 as per your concerns.

We also mentioned under the section 3.1. Nutritional status as the following:

“Responses to each question of the MNA are mentioned in the Supplementary material 1.” [PAGE 5, line 172-173]

Round 3

Reviewer 2 Report

Comments and Suggestions for Authors

The authors have addressed properly the previous cristicisms and although the confounding factors in the statistical analysis still exist and are unaccounted for, they were well described and the limitations of the study were suitably described. As such, the manuscript could be published as it is.